# Pitt-Hopkins Syndrome: Clinical and Molecular Findings of a 5-Year-Old Patient

**DOI:** 10.3390/genes11060596

**Published:** 2020-05-28

**Authors:** Florin Tripon, Alina Bogliș, Cristian Micheu, Ioana Streață, Claudia Bănescu

**Affiliations:** 1Laboratory of Medical Genetics, Emergency Clinical County Hospital Târgu Mureș, 540136 Târgu Mureș, Romania; florin.tripon@umfst.ro (F.T.); claudia.banescu@umfst.ro (C.B.); 2Department of Genetics, George Emil Palade University of Medicine, Pharmacy, Science, and Technology of Târgu Mureș, 540142 Târgu Mureș, Romania; 3Laboratory of Molecular Biology/Genetics, Center for Advanced Medical and Pharmaceutical Research, George Emil Palade University of Medicine, Pharmacy, Science, and Technology of Târgu Mureș, 540142 Târgu Mureș, Romania; 4Child Neurology Psychiatry Clinic, Clinical County Hospital Mureș, 540072 Târgu Mureş, Romania; cristianmicheu@yahoo.com; 5Regional Center for Medical Genetics Dolj—Clinical County Emergency Hospital Craiova, University of Medicine and Pharmacy Craiova, 200642 Craiova, Romania; ioana.streata@geneticamedicala.ro

**Keywords:** Pitt-Hopkins syndrome, *TCF4* deletion, neurodevelopment disorder, Face2Gene

## Abstract

Pitt Hopkins syndrome (PTHS) is a very rare condition and until now, approximately 500 patients were reported worldwide, of which not all are genetically confirmed. Usually, individuals with variants affecting exons 1 to 5 in the *TCF4* gene associate mild intellectual disability (ID), between exons 5 to 8, moderate to severe ID and sometimes have some of the characteristics of PTHS, and variants starting from exon 9 to exon 20 associate a typical PTHS phenotype. In this report, we describe the clinical and molecular findings of a Caucasian boy diagnosed with PTHS. PTHS phenotype is described including craniofacial dysmorphism with brachycephaly, biparietal narrowing, wide nasal bridge, thin and linear lateral eyebrows, palpebral edema, full cheeks, short philtrum, wide mouth with prominent and everted lips, prominent Cupid’s bow, downturned corners of the mouth, microdontia and also the clinical management of the patient. The previously and the current diagnosis scores are described in this report and also the challenges and their benefits for an accurate and early diagnosis.

## 1. Introduction

Pitt Hopkins syndrome (PTHS MIM #610954) is a neurodevelopment disorder associated with syndromic intellectual disability, cognitive and behavioral symptoms, and facial dysmorphism [1]. Marangi et al. and Whalen et al. [2,3] published two sets of criteria for the clinical diagnosis. Currently, the criteria for the clinical diagnosis were redefined in the first consensus statement [1] due to the fact that the two sets published before were not sufficiently precise [1,4]. As a result, signs and symptoms such as severe intellectual disability, limited or loss of speech, delay in gross motor skills, affected intestinal motility (constipation), dysregulated respiration (intermittent hyperventilation and/or apnea), absence of congenital malformations, distinctive facial appearance including a narrow forehead, thin lateral eyebrows, specific nose conformation with a broad nasal bridge, ridge and a bulbous tip with flared nasal alae, full cheeks/prominent midface, and wide mouth/full lips/cupid bow upper lip, etc., were considered suggestive for PTHS in the latest criteria for clinical diagnosis (moreover, some of the signs and symptoms have been described and used previously) [1,2,3,5,6,7,8]. From a clinical point of view, even if the latest criteria for the clinical diagnosis are more specific, the fact that there are some overlapping signs and symptoms between PTHS and Angelman, Mowat-Wilson, Kleefstra, Rett, etc., syndromes, represents a challenge for physicians to recognize and diagnose PTHS [1,8] especially in younger patients where the clinical diagnosis may be extremely difficult in some situations [1]. Another contribution to the challenge of clinical diagnosis is the fact that PTHS is a very rare condition, and until now, according to the Genetics Home Reference database, approximately 500 patients have been reported worldwide, not all of which are genetically confirmed.

Interstitial deletions, one partial duplication, point mutations, and other structural chromosomal abnormalities [1,7,9] of the Transcription Factor 4 (*TCF4*) gene were reported to be pathogenic for PTHS. Functional studies have previously demonstrated the fact that the dysfunction of the TCF4 protein codified by the *TCF4* gene explains the symptomatology of PTHS [10,11,12], and one of them suggested the role of SCN10a as a potential therapeutic target for PTHS [12]. The *TCF4* gene includes 20 exons. According to the latest international consensus [1], individuals with variants affecting exons 1 to 5 in the *TCF4* gene have mild intellectual disability (ID), between exons 5 to 8, moderate to severe ID and sometimes have some of the characteristics of PTHS, and variants starting from exon 9 to exon 20 have a typical PTHS phenotype [1].

Herein, we describe the clinical and molecular findings of a Caucasian boy diagnosed with PTHS at the age of 5 years, due to a large deletion, including the *TCF4* gene.

This study was approved by the Ethics Committee of the George Emil Palade University of Medicine, Pharmacy, Science, and Technology of Târgu Mureș, Romania (No. 66/27 April 2018). The parents of the patient provided informed consent for genetic testing and publication.

## 2. Case Report

### 2.1. Presenting Concerns

The patient was referred to our Genetics Department of the Emergency County Hospital from Târgu Mureș, Romania, for severe global developmental delay, intellectual disability, sensorineural hearing loss, epilepsy with an onset at the age of four years, under medication, and dysmorphic features. The patient, the third child of a nonconsanguineous marriage, was born from an uncomplicated pregnancy via vaginal delivery, at term and cranial presentation, with a birth weight of 3100 g, length of 52 cm, and an Appearance, Pulse, Grimace, Activity, Respiration (APGAR) score of 9 at the first minute and 10 at five minutes. He also presented a poor weight gain, chronic constipation, severe generalized hypotonia, stereotypic movements of the hands (hand wringing) with repetitive self-injurious behavior (self-biting), and language impairment. The karyotype was performed before and was normal.

### 2.2. Clinical Findings

At the moment of clinical genetic examination, the patient presented the following phenotypic traits: craniofacial dysmorphism with brachycephaly, biparietal narrowing, wide nasal bridge, thin and linear lateral eyebrows, palpebral edema, full cheeks, short philtrum, wide mouth with prominent and everted lips, prominent Cupid’s bow, downturned corners of the mouth, microdontia (Figure 1), bilateral single palmar crease, clinodactyly, thoracolumbar scoliosis, skin paleness, paraumbilical café au lait spot of 1/0.5 cm, cryptorchidism, hypopigmented scrotum, and severe hypotonia (the patient was unable to walk and had severe speech difficulties).

### 2.3. Diagnostic Focus and Assessment

The initial transfontanelle ultrasonography showed ventriculomegaly, and a computed tomography (CT) scan revealed cerebral atrophy and hydrocephaly. The abdominal ultrasound showed left hydronephrosis. A scrotal ultrasound was performed to determine the location of cryptorchid testicles and found both testicles within the inguinal canals. The echocardiography showed a patent foramen ovale with good ventricular contractility. Besides the sensorineural hearing loss, the oto-rhino-laryngology assessment also found congenital laryngeal stridor and chronic otitis media. Based on the patient’s phenotype and clinical features, we suspected PTHS.

Based on this suspicion, we evaluated the clinical diagnosis criteria at the first clinical presentation of the patient, using the previous criteria [2,3] and the current criteria for PTHS [1]; scores of 14 points and 8 points, respectively, were revealed. Based on the clinical phenotype of the patient and taking into consideration the scores that warrant *TCF4* molecular analysis, we decided to perform the genetic investigation immediately.

Multiplex Ligation-dependent Probe Amplification (MLPA) probemixes SALSA MLPA P297 and SALSA MLPA P075 probemixes (MRC Holland, Amsterdam, The Netherlands) were available in our laboratory, and firstly we performed MLPA analysis with the mentioned kits, according to the protocol recommended by the manufacturer. DNA was isolated from peripheral leucocytes using the Wizard^®^ Genomic DNA Purification Kit (Promega, Madison, WI, USA). A 3500 Genetic Analyzer (Applied Biosystems, Foster City, CA, USA) was used for capillary electrophoresis according to the MLPA protocol. For copy number change (CNC) detection from DNA of the patient, Coffalyser software (MRC-Holland) was used together with four reference DNA samples without CNCs. The SALSA MLPA P297 probemix contains two probes for the *TCF4* gene (covering exon 1 and 20) while the SALSA MLPA P075 used for results confirmation contains 35 probes for the *TCF4* gene (covering all exons of the *TCF4* gene).

The MLPA analysis performed with the P297 probemix revealed one signal suggestive of a heterozygous deletion (corresponding to exon 1) based on the two probes included. The analysis performed with the P075 probemix confirmed the heterozygous deletion of exons 1 to 8 of the *TCF4* gene. The other exons of the gene, namely, exons 9 to 20, were not deleted; the signals obtained were in normal ranges (Figure 2).

In order to determinate the breakpoints of the deletion, to find out if these deletions involved other genes that could contribute to the clinical phenotype of the patient and to find out if the patient does have other *TCF4* pathogenic variants, Microarray-based Comparative Genomic Hybridization (aCGH) and next-generation sequencing (NGS) analyses were carried out.

High-resolution microarray (aCGH) analysis was performed with CytoSure ISCA V2 CGH 8 × 60K microarrays, using commercially available male and female genomic DNA (Promega, Human Reference DNA, male and female). All the procedures (fluorescent labeling, purification of both patient and reference gDNA, hybridization, microarray wash, and microarray scanning and analysis) were performed according to the manufacturer’s protocol (Cytosure protocol—Version 3: May 2015). Array images were obtained using the NimbleGen MS 200 Microarray Scanner (Roche). Agilent Feature extraction software was used for data extraction. Copy number data were analyzed with CytoSure Interpret software (Oxford Gene Technology). The aCGH analysis revealed a loss of genomic material corresponding to a deletion on the long arm of chromosome 18, the 18q21.2-q22.1 region, of approximately 8.64 Mb: Arr[GRCh37] 18q21.2–q22.1(52961927–61602424)x1 (ISCN 2016). The deleted region encompasses 57 RefSeq genes, of which 36 are OMIM genes: *TCF4*, *TXNL1*, *WDR7*, *LINC-ROR*, *ST8SIA3*, *ONECUT2*, *FECH*, *NARS*, *ATP8B1*, *NEDD4L*, *MIR122*, *MALT1*, *GRP*, *RAX*, *CPLX4*, *LMAN1*, *CCBE1*, *PMAIP1*, *MC4R*, *CDH20*, *RNF152*, *PIGN*, *TNFRSF11A*, *PHLPP1*, *BCL2*, *KDSR*, *VPS4B*, *SERPINB5*, *SERPINB12*, *SERPINB13*, *SERPINB4*, *SERPINB3*, and *SERPINB11*.

NGS analysis was performed using an Epilepsy Panel (including Preliminary-evidence Genes for Epilepsy), which included 181 genes (Invitae, San Francisco, CA, USA). CNC analysis of the sequenced genes was also performed. The NGS analysis revealed three deletions: a heterozygous deletion of exons 1 to 8 of the *TCF4* gene (18q21.2) considered pathogenic (associated with autosomal dominant PTHS) a heterozygous deletion of the entire coding sequence of the *PIGN* gene (18q21.33) considered pathogenic (associated with the autosomal recessive PIGN-congenital disorder of glycosylation), and a heterozygous deletion of the entire coding sequence of the *NEDD4L* gene (18q21.31) considered a variant with uncertain significance.

The parents of the child were investigated by MLPA using the P075 probemix, but the results were in normal ranges, suggesting “the novo” appearance of the child deletion. Genetic counseling was offered to the patient’s family, and we also explained the recurrence risk of 2% and the possibility of prenatal testing in the case of a new pregnancy.

### 2.4. Facial Analysis Technology

In order to test the utility of the facial analysis technology for PTHS, we used the CLINIC application of the Face2Gene (web interface, version 19.1.3; www.face2gene.com). Face2Gene uses deep-learning algorithms and it has been reported to be a useful technology for some rare diseases, mainly for pediatricians, general practitioners, but also for the young geneticist. The results may be interpreted with caution because more studies are needed to understand the genotype–phenotype correlations [13,14]. In the case of our patient, based on gestalt comparison, the score was high and suggested the possibility of a patient with PTHS (Figure 3).

### 2.5. Therapeutic Focus and Assessment

Sodium valproate (Depakine) syrup 3 × 2.5 mL/day, Nitrazepam 2.5 mg tablets 3 × 1.5/day, Glycerol (50%) solution 3 × 5 mL/day, and Phenobarbital tablets 3 × 0.25/day were administrated daily. Alternatively, Cerebrolysin 2 mL/day intra-muscular, Vitamin B12 50 micrograms/day, and phosphothreonine (Tonotil N) 2 × 2.5 mL/day for 10 days, followed by 20 days with Pyritinol (Encephabol) solution 2 × 5 mL/day, Tanakan solution 2 × 1 mL/day, and Levocarnitinum (Carnil) solution 100 mg/mL, 2 × 5 mL/day were administrated. Kinetotherapy, electrotherapy, play therapy, medical-gymnastics, psychic stimulation, and speech development therapy were also recommended and performed.

### 2.6. Follow-Up and Outcome

The child was periodically reevaluated by physicians specialized in Pediatrics, Child Neurology and Psychiatry, Medical Genetics, and also by a kinetotherapist and psychologist for medical-gymnastics, psychic stimulation, and speech development therapy (the mother respects all recommendations at home, and she has been trained to perform, as possible, the medical-gymnastics, psychic stimulation, and speech development therapy). The patient was also referred to Pediatric Orthopedics and Ophthalmology for specialist examination. Special orthotics and a corset were recommended and made for the child. A slow reduction of hydronephrosis was observed, currently requiring ultrasound monitorization. Under epileptic medication, the seizures did not reappear.

Despite our management, currently, the child has severe hypotonia, can walk only if supported, and has severe speech difficulties. Five months after our first evaluation, the child developed breathing regulation anomalies (hyperventilation and apnea) shortly after he fell asleep. According to the mother’s account, these anomalies appeared mostly when the child was tired. She would wake him up by gentle movements, after which he would breathe normally and fall back asleep immediately, without the breathing regulation anomaly reappearing. Until now, the polysomnography and surgical treatment of cryptorchidism were not performed despite the recommendation of the physician.

## 3. Discussion

In the present study, we describe the clinical features and molecular results of a Caucasian boy with PTHS.

Several variants of the *TCF4* gene were described, mostly between exons 7 to 20. According to Ensembl Genome Browser, *TCF4* pathogenic variants are associated with the autosomal dominant non-syndromic ID, PTHS, while our patient’s phenotype and clinical features are suggestive of PTHS.

Based on aCGH results, similar cases with large cryptic deletions were reported by Giurgea I et al. [15]. Similar to our affected child, the patients reported by Giurgea I et al. [15] presented severe mental retardation, speech difficulties, hypotonia (unable to walk), and stereotypic movements of hands. Our case also presented facial dysmorphism (Figure 1), epilepsy, breathing regulation anomalies, and sensorineural hearing loss. However, based on MLPA and NGS analysis, our patient had deletion only of exons 1 to 8 of the *TCF4* gene, suggesting that aCGH resolution was not sufficiently high to reveal the precise breakpoints between exons 8 and 9 of the *TCF4* gene.

Among the *TCF4* variants discussed in the report of Bedeschi et al. [16], those associated with a severe but atypical PTHS phenotype occurred within exons 7 and 8, with alternative splicing being preserved.

Previous studies [17,18] demonstrated the complexity of the *TCF4* gene structure and the alternative splicing of this gene expressed highly in the nervous system. Several alternative transcripts were described by Timmunsk et al. [18] of which two TCF4 protein isoforms are more important, TCF4 long isoform B (TCF4-B) and short isoform A (TCF4-A). The deletion of exons 1 to 8 of the *TCF4* gene, identified in our patient, will disrupt TCF4-B and will leave the TCF4-A isoform intact [3,17,18]. As a consequence, the codified TCF4 protein will be shorter, i.e., the short (TCF4S) isoform [19]. The preservation of the TCF4S isoform, caused by deletions or very rare mutations in the proximal part of the complex *TCF4* gene, will influence the transcription activation (lower nuclear enrichment and reduced transcriptional activity), this being responsible for the milder phenotype of PTHS patients [18,19]. The lack of longer isoforms may upregulate other genes involved in the development, or the shorter isoforms might be overexpressed that might lead to mild ID [20].

The patient described in our study had deleted transactivation domains (AD1) and nuclear localization signal (NLS), with the AD2 domain and basic helix-loop-helix (bHLH) being normal. Based on MLPA ligation sites, the promoter for TCF4-A, in the case of our patient, was not included in the deletion. The recent study published by Wedel M et al. [19] described the effect of such a deletion that generates a TCF4S isoform and has a “dramatic effect on oligodendroglial development” and central nervous system myelination [19]. Even if the impact of this deletion was previously described, unfortunately for our patient, we were unable to analyze the *TCF4* transcripts or protein due to financial limitations. The parents refused further blood harvest for cDNA and protein analysis.

Usually, according to the first PTHS consensus [1], variants of exons 9 to 20 are associated with a typical phenotype of PTHS, but exceptions may exist, for example, one was reported by Tan et al. [21], who described a unique case of a Caucasian male with PTHS caused by a single-pair deletion in exon 19 of the *TCF4* gene (c.1933delG) with mild to moderate ID and independent ambulation [21].

Even if PTHS is a very rare syndrome, the positive diagnosis can be revealed by the specific phenotype and the clinical features described. There may be insufficient awareness of PTHS, and new strategies and education programs are needed for clinical diagnosis and management. In this way, more physicians will recognize the clinical phenotype of PTHS, and referral of patients for genetic confirmation will lead to an accurate diagnosis and an increased and more precise genetic database. Facial analysis technology may be a promising tool, but future studies are needed to demonstrate its accuracy. However, advances in PTHS were made starting with the recent international consensus publication and the realization of the fact that continuous reevaluation of the consensus is needed [1].

Regarding the molecular technique used on our PTHS patient, we admit that MLPA represents a fast multiplex technique that is useful and cost-effective, likely being the most efficient method for the detection of partial gene deletions (and duplications), but it does not allow for the precise characterization of the molecular defect [22,23,24,25]. NGS and aCGH analysis have shown that the genetic and phenotypic heterogeneity of patients with intellectual disabilities is very high. This would favor the use of genome-wide tests instead of targeted testing. We recognize that the financial possibilities and lack of infrastructure in low or low–middle income countries are difficulties that interfere with the implementation of such tests, but the price is decreasing, and new products are being developed [3].

The phenotype of our patient was suggestive for PTHS, being similar to that reported in the literature previously [5,6,7,26]. We recommend the periodic reevaluation of PTHS patients without genetic confirmation and especially of PTH-like patients or patients where the clinical diagnosis is not precise, with a clinical diagnostic score ≤ 8 [1], suggesting possible PTHS. This is recommended because in our case, during subsequent reevaluation and recalculation of the clinical diagnostic score, when adding breathing regulation anomalies, the score increased (10 points) and showed clear indication for molecular analysis. The fact that we performed the molecular analysis based on the clinical phenotype at the first evaluation, with an 8-point score [1], led to an early diagnosis and appropriate genetic counseling for the family. Thus, our report may be considered a confirmation of the recently reported clinical diagnostic criteria for PTHS [1] and importantly, as further evidence of the strict connection between clinics and the laboratory for precise genetic diagnosis.

Facial analysis technology, in the case of PTHS in this study, proved to be promising technology that may be useful for clinicians. Before being implemented as an additional tool for clinical diagnosis, future studies are required to demonstrate its accuracy in practice.

The particularity of our patient represents the fact that the patient was diagnosed with a very rare syndrome due to a large deletion (18q21.2–q22.1 region, including exons 1 to 8 of the *TCF4* gene) and severe phenotype. In addition, our study underlines that a clinical diagnosis of PTHS can be made based on the phenotype and the recently reported clinical diagnostic criteria for PTHS [1]. Confirmation may be done by sequencing (NGS), MLPA of the *TCF4* gene, aCGH, and sometimes by karyotype (to reveal balanced chromosome rearrangements that disrupt the *TCF4* gene).

## 4. Conclusions

In conclusion, Pitt-Hopkins syndrome should be considered in children with specific craniofacial dysmorphism, severe developmental delay, intellectual disability, breathing anomalies, and disturbances of intestinal motility. The diagnosis of PTHS is a clinical one; according to the clinical diagnostic scores, periodic reevaluation and genetic confirmation must be performed, sometimes with several necessary techniques due to the multiple variants that can be found simultaneously. Our report demonstrated the utility of the recently reported clinical diagnostic criteria for PTHS and the facial analysis technology for PTHS and highlighted the strict connection between clinics and the laboratory for precise genetic diagnosis.

## Figures and Tables

**Figure 1 genes-11-00596-f001:**
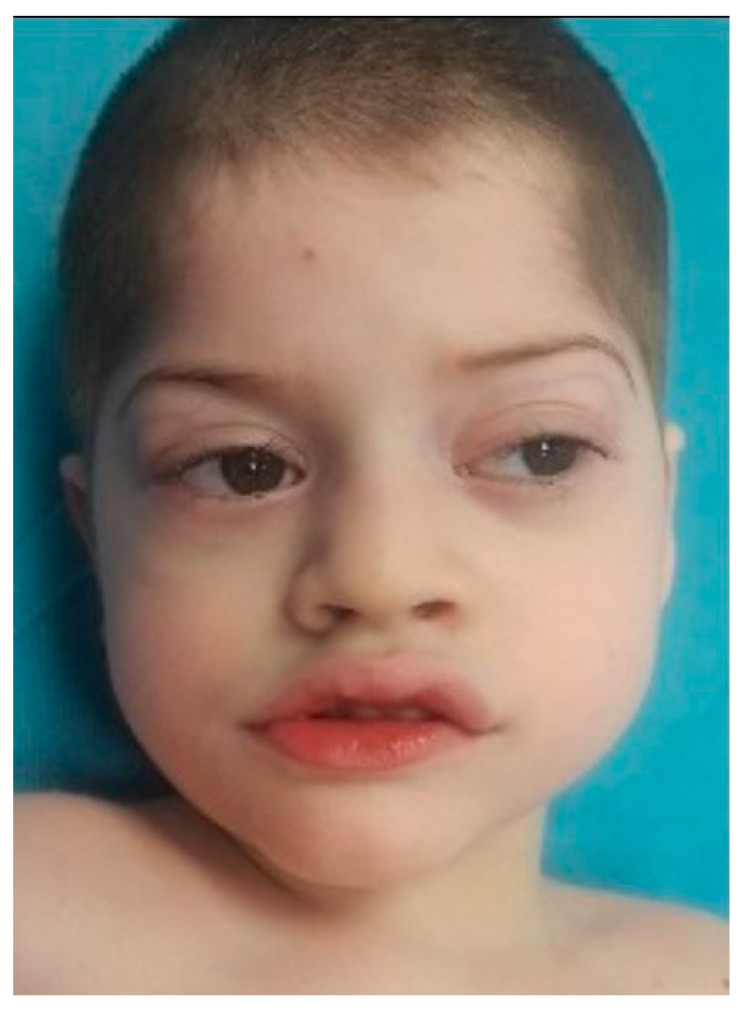
Facial dysmorphism in a 5-year-old boy with Pitt Hopkins syndrome (at the time of first genetic evaluation). Written informed consent from the parents of the patient was obtained for the publication of this image.

**Figure 2 genes-11-00596-f002:**
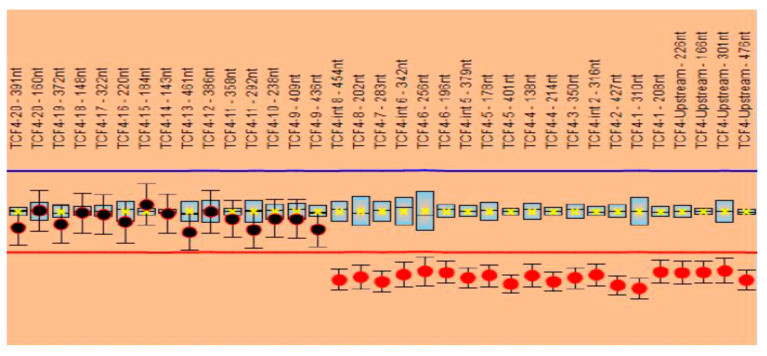
Salsa MLPA P075 results. Heterozygous deletion of exons 1 to 8 and normal ranges for exons 9 to 20.

**Figure 3 genes-11-00596-f003:**
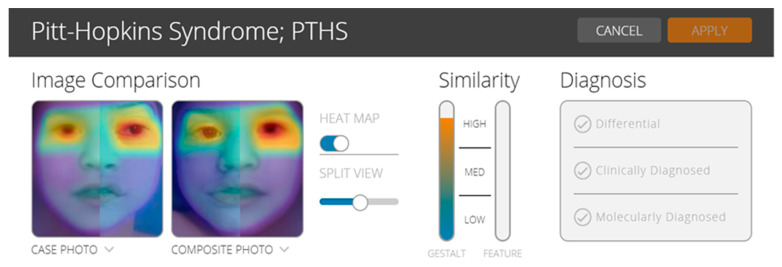
Face2Gene analysis based on a gestalt comparison. Personal archive.

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
