# Peer review of "Pitt-Hopkins Syndrome: Clinical and Molecular Findings of a 5-Year-Old Patient"

_genes, 2020, doi:10.3390/genes11060596_

Round 1

Reviewer 1 Report

Authors report on a typical PTHS patient with TCF4 haploinsufficiency due to a relatively large chromosome deletion including exons 1-8 of TCF4.

Discussion about intragenic genotype-phenotype correlations is inappropriate.

As mentioned by the Authors, there is literature evidence that gene variants affecting exon 8 are associated to a severe neurodevelopmental disorder in the clinical spetrum of PTHS but to a less consistent facial phenotype. However it must be specified that these specific gene variants are point mutations, leaving alterantive splicing of intermediate or short transcripts preserved. That is not the case for a complete deletion of exons 1-8.

The present report does not add new insights in the molecular pathogenesis of PTHS. On the contrary it could generate confusion about nosology of this condition. 

Reviewer 2 Report

The authors of this report describe a single patient with suspected Pitt-Hopkins syndrome (PTHS) with a contiguous gene deletion that encompasses part the TCF4 gene on chromosome 18. The MPLA results seem relatively clear and indicate that the deletion only encompasses exons 1-8. The "full" presentation of PTHS in this patient is unusual where milder phenotypes are reported to be the norm.  The authors appear to have missed an important subtlety in the patient - the deletion will disrupt TCF4B but should leave TCF4A intact (for a review article see PMID:24594265). The preservation of the shorter TCF4 isoforms are thought to be responsible for the milder present of PTHS patients with deletions or very rare mutations in the proximal part of the complex TCF4 gene. The authors should reference the important work from the Timmusk lab that demonstrated the complexity of the TCF4 gene (PMID:21789225). The molecular papers on TCF4 are more relevant to this paper than the studies in mouse that largely ignore the direct molecular function of TCF4. The authors are also encouraged to read the most recent paper on a new TCF4 model lacking only TCF4B that was published a few weeks ago (PMID:32266943). Given that the MPLA has low resolution it would be useful to show the absence or presence of the TCF4 transcripts or protein in this patient. PCR on RNA extracted from white blood cells in this case compared to a control(s) would make an important addition to this paper. Is the TCF4A promoter included in the deletion? I am not convinced that patient's image should form part of this paper.

Round 2

Reviewer 1 Report

I read with attention the present revised version. However I have a couple of majors criticisms, which must be satisfied before considering the MS suitable for publication"

1) Introduction, line 22. According the the cited guidelines for ther diagnosis and management of PTHS patients (Clin Genet, 2019), the statement "deletions between exons 5 to 8 are associated to a PTH-like syndrome phenotype" is incomplete and it sounds incorrect. Authors have to report the proper statement: "individuals with variants affecting exons 5 to 8 present with moderate to severe ID and sometimes have some of the characteristics of PTHS". First, this statement is about intragenic variants, and not about large chromosome deletions, as diagnosed in the present patients. Although Authors spent a lot of additional work in discussing about alternative gene transcripts, they did not perform cDNA analysis, neither investigations at a protein level, as they comment on. These further investigations are not requested by the present reviewer, however Authors must discuss about that in a more clear manner.

2) The second criticism, and the most important one, is about the statement on page 3, line 95, about the clinical score of their patient that, according to the above mentioned guidelines, was of 8: "Despite that, we decided to perform the genetic investigation immediately" This statement is completely wrong, and it can create confusion in the scientific community. In the cited literature paper, it is clearly indicated that "A score of 6-8 warrant TCF4 molecular analysis". Authors are kindly invited to modify the test accordingly.

A final comment is in general about PTHS nosology. This single case report can finally deserve publication, since it is well described. However it should be reported as a confirmation of the recently reported clinical diagnostic criteria for PTHS, and, importantly, as a further evidence of the strict connection between clinics and laboratory for the precision genetic diagnosis.

Reviewer 2 Report

Without independent verification using a different technique the authors conclusions are not entirely solid. Addition of new phenotyping data was not requested so I am at a loss to why the authors have included such data. As it stands this is merely a case report of a single patient with PTH syndrome and may be more suited to a specialized journal that accepts this type of report.
